# Comparison of REST and GraphQL Interfaces for OPC UA

**Riku Ala-Laurinaho** [1,*], **Joel Mattila** [1], **Juuso Autiosalo** [1], **Jani Hietala** [2], **Heikki Laaki** [1] **and Kari Tammi** [1]

[1] Department of Mechanical Engineering, Aalto University, 02150 Espoo, Finland; joel.mattila@aalto.fi (J.M.); juuso.autiosalo@aalto.fi (J.A.); heikki.laaki@aalto.fi (H.L.); kari.tammi@aalto.fi (K.T.)
[2] VTT Technical Research Centre of Finland Ltd., 02044 Espoo, Finland; jani.hietala@vtt.fi
* Correspondence: riku.ala-laurinaho@aalto.fi

**Abstract:** Industry 4.0 and Cyber-physical systems require easy access to shop-floor data, which allows the monitoring and optimization of the manufacturing process. To achieve this, several papers have proposed various ways to make OPC UA (Open Platform Communications Unified Architecture), a standard protocol for industrial communication, RESTful (Representational State Transfer). As an alternative to REST, GraphQL has recently gained popularity amongst web developers. This paper compares the characteristics of the REST and GraphQL interfaces for OPC UA and conducts measurements on reading and writing data. The measurements show that GraphQL offers better performance than REST when multiple values are read or written, whereas REST is faster with single values. However, using OPC UA directly outperforms both REST and GraphQL interfaces. As a conclusion, this paper recommends using a GraphQL interface alongside an OPC UA server in smart factories to simultaneously yield easy data access, the best performance, and maximum interoperability.

**Keywords:** communication; GraphQL; Industry 4.0; interfaces; OPC UA; REST

## 1. Introduction

The need for easier data access and faster development drives a shift towards web-based technologies in industry. Web technologies enhance interoperability and allow communication between various entities in cyber-physical systems and smart factories. OPC UA (Open Platform Communications Unified Architecture), a commonly used standard for industrial communication [1], attempts to merge traditional industrial communication with modern web technologies [2]. It is platform-independent, provides an information and communication model, and allows interoperability between various entities from sensors to Enterprise Resource Planning (ERP) systems [3]. An OPC UA server can be used to provide access to a machine PLC (Programmable Logic Controller) system turning it into an IIoT (Industrial Internet of Things) device. In addition, it allows data to be collected from field devices for advanced applications, such as optimization, analysis, and predictive maintenance.

However, OPC UA has some disadvantages compared to native web technologies: several round trips are needed to establish a connection before data can be accessed (Figure 1) [2]; it is more complex, making application development laborious; and there is a need for a specific client. Lack of client-side libraries especially hinders application development. An example of an industrial application in which a GraphQL interface allowed easier access to an overhead crane OPC UA server is presented in [4]. To overcome the disadvantages and limitations of OPC UA, several papers have proposed making OPC UA RESTful (Representational State Transfer) [1,2,5,6]. Recently, a GraphQL interface for OPC UA servers has also been proposed in the previous work by these authors [7,8]. GraphQL is an emerging approach to building web APIs (Application Programming Interfaces) designed to address issues with REST APIs; these issues include overfetching

and multiple requests needed to fetch desired data. This paper builds upon the work presented in [7,8] and compares GraphQL and REST in industrial communication with OPC UA to bring benefits of web interfaces, such as developer-friendliness and interoperability with almost any web-capable device using HTTP (Hypertext Transfer Protocol), to industrial domain. Even though OPC UA can also be accessed via HTTP [3], which increases the interoperability, this access method might not be supported by the server. The scientific literature comparing REST and GraphQL with OPC UA and in industrial communication is scarce, and the main contributions of this paper are as follows:

1.  Analyzing the characteristics of REST and GraphQL interfaces for OPC UA servers.
2.  Comparing the performance of REST and GraphQL interfaces.
3.  Promoting easier shop floor data access and interoperability for Industry 4.0 with HTTP-based web interfaces.

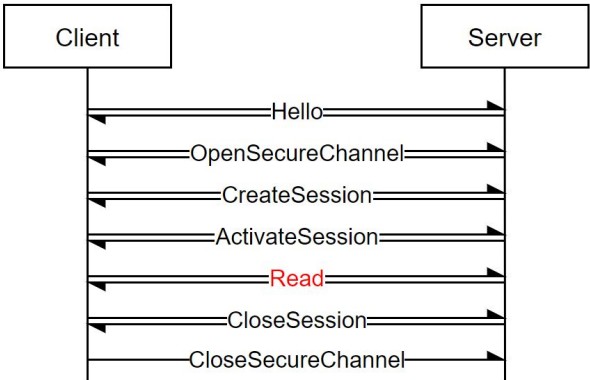

**Figure 1.** OPC UA (Open Platform Communications Unified Architecture) requires several round trips before data can be read from a server [2].

## 2. Background

### 2.1. OPC UA

OPC UA is a standard for industrial communication, which promotes the interoperability of systems [3]. It is platform-independent and suitable for various devices, from embedded systems to the cloud [9], allowing information exchange between these entities [3]. The standard is in constant development by the OPC UA Foundation [2], as evidenced by four extensions to the standard being published during the year 2020 alone.

OPC UA offers both a client–server and publish–subscribe model for communication [3]. In the client–server model, a server offers services that clients can use to interact with the server and its nodes [2]. A server may provide only a subset of these services, which are grouped into *Service Sets*, such as the *NodeManagement* and *Query Service Set* [3]. Services can be requested over OPC UA TCP (Transmission Control Protocol) using OPC UA Binary, XML (Extensible Markup Language), or JSON (JavaScript Object Notation) data formats [3]. The publish–subscribe model is favored for many-to-many communication in which data is distributed from several publishers to several subscribers [9]. This model can be implemented with a broker-less approach using a network infrastructure as Message Oriented Middleware (MOM) or using the broker as MOM with publish–subscribe protocols, such as MQTT (Message Queuing Telemetry Transport) or AMQP (Advanced Message Queuing Protocol) [3].

The OPC UA information model consists of nodes and references between them, forming a graph data structure [2]. Nodes describe real-world objects, their properties, and their relationships [10]. Each node belongs to a certain *NodeClass* and its attributes are defined by this class [1]. Nodes also have a few common attributes such as *NodeId* and *DisplayName*. The relationships between nodes can be divided into *HierarchicalReferences* and *NonHierarchicalReferences* [11]. A set of nodes, which an OPC UA server makes available to a client, forms an AddressSpace [3]. OPC UA also provides Companion Specifications

which define the information model and its semantics for a certain domain allowing interoperability [2].

*2.2. REST*

REST was originally developed as design guidelines for distributed hypermedia systems, such as the World-Wide Web, by Fielding [12]. These guidelines were intended to describe the design principles of the Web (which were not comprehensively documented at the time) and improve its architecture, for example, by enhancing scalability. The architectural constraints of REST according to Fielding [12] are as follows:

1.  **The Client–Server** model aims to separate concerns: the client is responsible for the user interface whereas the server stores data and offers services to clients. This model allows better scalability by simplifying the server, and the development of client and server can be separately conducted.
2.  **Stateless** communication does not allow the server to store any information about the client, and the client is responsible for maintaining the session state. Therefore, "each request from client to server must contain all of the information necessary to understand the request". Stateless communication improves scalability, but increases the repetitive data sent with requests.
3.  **Cache** allows responding to identical requests with the same stored response. Cache reduces network consumption, decreases latency, and improves scalability. When using cached responses, the data freshness needs to be ensured, for example, with *Cache-Control* header.
4.  **Uniform interface** allows a simpler system architecture and separation of the implementation of components from the services they offer. It induces a few additional requirements into the interface design: "identification of resources, manipulation of resources through representations, self-descriptive messages, and, hypermedia as the engine of application state" (HATEOAS).
5.  **Layered system** consists of hierarchical layers, and the components on each of these layers are only able to directly interact with components on the immediate layers. Layered architecture reduces the system complexity, but adds latency and overhead. Intermediary components can be used for load balancing and transforming the content of messages.
6.  **Code-On-Demand** enables a client to download and execute code from the server, extending the client capabilities. This is the only optional constraint.

REST is the de facto architectural style for building web APIs. It is also commonly misused as a synonym for any HTTP API, and a mobile data analysis by Rodríguez et al. [13] showed that very few of the so-called REST APIs fully comply with the architectural constraints. REST is not a standardized architecture, and there are no explicitly defined rules on how it should be applied to HTTP APIs. Nevertheless, there are several established best practices for building RESTful HTTP APIs, the most important of which are as follows:

1.  Information is organized into resources and each resource is identified by a URI (Uniform Resource Identifier) [14].
2.  HTTP methods (POST, GET, PUT/(PATCH), DELETE) are used for CRUD (Create, Read, Update, Delete) operations [15].
3.  Consistent use of HTTP status codes, such as responding *201 Created* to POST request after successful creation of a resource [15].
4.  HATEOAS. Each resource representation contains links to related resources with possible operations to them [14]. There can be multiple representations of a resource separated from the storage method on the server, and the representation type can be indicated in the headers of the response [13]. It should be possible to browse through the API without any previous knowledge of the API structure by following links [14].

*2.3. RESTful OPC UA*

　　To improve the interoperability of OPC UA and provide easier data access, it has been proposed to make OPC UA RESTful [1,2,6]. RESTful communication would allow several benefits, such as removing the need for several handshakes to establish a connection to the OPC UA server (Figure 1), providing better scalability, and improved cacheability. There are two approaches for making OPC UA RESTful: using gateways that offer REST interface translating requests from clients to OPC UA service requests [1,5]; and applying modifications to the OPC UA standard [2,6]. However, making OPC UA fully REST compliant has proven challenging because it is originally a stateful protocol, that is, the server stores information about clients (for example, sessions are needed for communication and several services require storing client information) [2,3]. In OPC UA specification version 1.04, a *SessionlessInvoke* service has been introduced to allow stateless communication [6,16]. However, only a limited number of services, such as Read service, are supported [16]. These services are consumed by embedding a service request into the body of the *SessionlessInvoke* request.

　　Grüner, Pfrommer and Palm [2] proposed extensions to OPC UA protocol to make it RESTful. The extensions included stateless communication without extra handshakes and caching with expiration tags. Stateless communication with OPC UA significantly reduced the request execution times. To further reduce communication overhead, Grüner et al. proposed OPC UA over UDP (User Datagram Protocol), which improved the performance compared to TCP at the expense of reliability.

　　Schiekofer, Scholz, and Weyrich [6] proposed additions to the OPC UA standard to allow RESTful communication. These additions were primarily related to SessionlessInvoke, some of which were merged with version 1.04 of the OPC UA specification. They introduced a mapping of HTTP methods to the *SessionlessInvoke* (whereas, in OPC UA specification, only POST is used) as well as a description of resource representation by using MIME-types (Multipurpose Internet Mail Extensions), which allowed showing OPC UA resources on a browser. In addition, they implemented batch support for their RESTful OPC UA prototype. Compared to the work by Grüner et al. [2], Schiekofer et al. factored in the possibility that *Namespace* and *ServerArray* may change between subsequent sessionless requests and a *NodeId* can become erroneous. To overcome this, the version of the used *NamespaceArray* and *ServerArray* is included in the new *urisVersion* field. Later, Schiekofer and Weyrich [17] also presented an implementation of group-subscriptions to RESTful OPC UA.

　　Paronen [5] implemented a system for monitoring IIoT devices, which consists of a web application that provides HMI (Human–Machine Interface) for the OPC UA server as well as a web service that acts as a proxy between the web application and the OPC UA server and offers a REST API. Paronen used SSE (Server-Sent Events) to push data from the server to the client, which allowed the implementation of subscriptions. To reduce requests, Paronen also embedded the information of target and source nodes into the node representation. However, his solution revealed certain drawbacks including hard-coded API endpoints and the need to store the client state on the web service server; thus resulting in the solution not being fully REST compliant.

　　Cavalieri, Salafia and Scroppo [1] implemented a RESTful web platform for accessing multiple OPC UA servers. The platform converts REST requests into OPC UA service requests. The use of the platform does not require any modifications to the OPC UA specification, and from an OPC UA server perspective, the web platform is a regular client. In addition to the RESTful interface, the platform also allows the monitoring of OPC UA objects with a subscription model, which is implemented with a separate broker. The platform is publicly available on GitHub [18].

　　Another server application that offers a REST interface for OPC UA servers is HyperUA [5] as mentioned in earlier publications [2,6]. However, it seems that HyperUA is no longer being developed and is not available for use.

Derhamy et al. [19] proposed a protocol translator for OPC UA to enhance the interoperability of IIoT. Their proposal extends the Arrowhead framework protocol translator service and allows using OPC UA with HTTP, CoAP (Constrained Application Protocol), and MQTT. The translator maps OPC UA services to CRUD operations, which are then mapped to HTTP/CoAP methods. Each OPC UA node is identified by URL, and, thus, the proposal includes some elements from REST architecture. The major drawback of the translator is the support for only 7 out of 37 OPC UA services.

### 2.4. GraphQL

GraphQL is a query language and a runtime for performing queries on a server [20]. The development of GraphQL was started at Facebook in 2012 when it was noticed that the current solutions for fetching data were not optimal for their mobile apps [21]. GraphQL was published as an open-source project in 2015 with the GraphQL Foundation leading its development since 2018, and, currently, several large companies have adopted GraphQL including Airbnb, GitHub, Netflix, and Twitter [22]. GraphQL is protocol agnostic, but in practice it is used with HTTP [23], similar to REST. In addition to the request–response model, GraphQL offers subscriptions allowing the server to push data to clients, which is often implemented with WebSockets [23].

GraphQL is an alternative to REST interface and has some major differences in a way the interface is used, especially in terms information model and HTTP methods. Contrary to REST, GraphQL queries are sent to a single endpoint, and HTTP methods do not have semantic meaning; in other words, both GET, in which the query is embedded into the URI, and POST requests, in which the query is in the body of the request, are used [24]. Data is modeled as a graph, and the structure is defined by a schema [25]. The schema defines the possible operations on data, which a GraphQL API server must provide [24]. The actual implementation of these operations is server-specific, and a GraphQL server can use any database or data storage [20]. In addition, the schema allows validation of the query [25] and, for application development, a schema-first approach, in which it is used to document the requirements for data helping front-end developers to communicate their needs to back-end developers [23]. The main design principles of GraphQL according to the specification [26] are as follows:

1.  **Hierarchical:** GraphQL queries are hierarchical, following the structure of the application. In addition, the query and the response are similar in form.
2.  **Product-centric:** GraphQL is built for the needs of applications and their front-end developers.
3.  **Strong-typing:** Data structure and types are explicitly defined, allowing the server to validate queries. This can also be utilized in the development phase since queries can be tested before they are implemented by a server.
4.  **Client-specified response:** a client defines exactly the data it wants within the query, and a server returns a response following the structure of the query.
5.  **Introspective:** The GraphQL type system can be queried, allowing its introspection. A browser-based Integrated Development Environment (IDE) called GraphiQL is provided for introspection of the schema and executing queries [23].

### 2.5. GraphQL for OPC UA

To bring the benefits of GraphQL to the industrial environment, Hietala [7] and Hietala et al. [8] developed a GraphQL wrapper for OPC UA. The wrapper translates a GraphQL query into one or more OPC UA service requests. For example, read requests can be batched into a single OPC UA service request, whereas writing a value or fetching a subnode requires a separate service request. GraphQL wrapper acts as a client for an OPC UA server, and several servers can be aggregated behind a single GraphQL wrapper, which allows access to the OPC UA server of each machine in a factory from a single endpoint and API. The wrapper is available as open-source software on GitHub [27].

In addition to the wrapper, authors have found two other open-source solutions for using OPC UA via GraphQL API. The first one is based on the Node.js Express web framework, and can be found from [28]. The second one uses Java and is called Frankenstein Automation Gateway [29]. It allows interaction with an OPC UA server also via MQTT. In the measurements, this paper relied on the GraphQL wrapper developed by the authors in [8].

### 3. Comparison of REST and GraphQL

Next, the characteristics of REST and GraphQL interfaces for OPC UA servers are compared in the context of industrial communication. The comparison is summarized in Table 1.

**Table 1.** Comparison between REST and GraphQL interfaces.

|  | REST | GraphQL |
|---|---|---|
| **Communication model** | Client-server | Client-server and Subscriptions |
| **Protocol** | HTTP [1] | HTTP [1] |
| **Cache** | At any point | Application-specific [2] |
| **Scalability** | Good | Medium |
| **Interface** | Uniform | Application-specific |
| **Ease of use/development** | Medium/Good | Good/Good [3] |
| **Bandwidth usage** | High | Low |
| **Performance** | Low | Medium |

[1] Not bound to a specific protocol, but used in practice with HTTP. [2] Implementing cache is up to the application developer. [3] Allows a schema-first approach [23].

#### 3.1. Communication Model

REST follows the client–server architecture [12] and is coupled with the request–response communication model. However, to bring subscriptions to RESTful OPC UA clients, Schiekofer and Weyrich presented a solution based on long-polling and ring buffers [17]. The ring buffers are necessary since REST is a stateless protocol and storing the state of the client, which is often required for the efficient implementation of subscriptions, is not allowed. Long-polling is needed because HTTP is a request–response protocol [30] and is not intended for publish–subscribe communication. Long-polling [31] is a technique in which the server does not immediately respond to a request but leaves the communication channel open ensuring that it can instantly return a response and effectively "push" data to the client when data becomes available. Even though the subscriptions could technically be implemented following REST (such as in [17]), the authors of this paper consider these solutions to be contradictory to the objectives of RESTful architectural style, such as scalability. This paper suggests that publish–subscribe communication with OPC UA should not be implemented following RESTful architecture.

On the other hand, GraphQL offers a request–response model and subscriptions that allow a client to request updates of certain objects and a server to push these updates to the client. In industrial communication, subscriptions might prove useful when multiple clients need updates from several servers, for example, when there are continuous status updates from multiple machines.

#### 3.2. Protocol

Both REST and GraphQL are protocol agnostic but are used with HTTP in practice. GraphQL uses GET and POST methods to transport queries, whereas REST uses GET, POST, PUT/PATCH, and DELETE methods, which are bound to CRUD operations. HTTP is the Internet protocol supported by all modern browsers. Due to its ubiquity and wide support, HTTP increases interoperability and the accessibility of data compared to OPC UA, which requires the use of an OPC UA specific client. (Some OPC UA servers might also support HTTP as a transport method.) The drawback of HTTP in industrial settings is its poor suitability for constrained devices due to its relatively large overhead and bandwidth

consumption, a higher amount of memory used, and more processing power required compared to other protocols [30]. GraphQL subscriptions, which require pushing data from server to client, can be implemented with, for example, WebSockets [23].

### 3.3. Cache

One major benefit of REST compared to GraphQL is the ease of caching. Because requests are sent to a specific URL, the cache can be linked to this unique identifier. REST allows caching on the client and server level as well as between them by using, for example, gateways and proxies [12]. Caching is also possible with GraphQL by creating globally unique identifiers for objects [32] or by using specific clients, such as Apollo, which handle caching by storing query results in the memory [23]. The GraphQL specification [26] does not define caching strategies, and it is up to the developer to implement caching for GraphQL.

In general, cache increases performance and reduces network traffic. However, in industrial settings, field device data may constantly change [6], thus preventing the use of cached values. OPC UA uses a *maxAge* parameter for read services to determine if the server can return a cached response [6]. For effective caching with OPC UA, REST and GraphQL APIs should utilize this parameter.

### 3.4. Scalability

Scalability means the server's ability to handle an increasing number of clients. Scalability is mainly determined by the resources needed for the communication with an individual client, the ability to use load-balancing servers, which entails sharing the workload [2], and caching. REST was intended to improve the scalability of Web with scalability being an integral part of it [12]. Scalability of REST and the use of load-balancing were some of the main motivators for Grüner, Pfrommer and Palm to make OPC UA RESTful [2]. The scalability of GraphQL is in theory poorer than REST because caching is more difficult. In addition, the possible subscriptions are not stateless and the connection to the client must be kept open, thus reserving resources. Nevertheless, the experiments without caching by Heredia, Flores-García, and Solano [33] indicate that GraphQL performs better than REST with multiple clients.

The scalability needed from an OPC UA server is highly dependent on the use case and field devices. For example, one machine may need to communicate with only a few adjacent machines. On the other hand, Grüner, Pfrommer and Palm provide an example of a use case requiring high scalability in which thousands of pallets in a storage periodically send their statuses [2]. In case of an emergency, such as a sudden temperature drop, these pallets may simultaneously send thousands of alert messages.

### 3.5. Uniform Interface

A distinguishing feature of the REST architectural style is its uniform interface. In practice, uniform interface means that with HTTP APIs, resources are identified with URIs, each HTTP method is bound to a certain type of operation, status codes indicate the result of operation, content negotiation can be used to retrieve a certain representation of a resource, the representation type is indicated by headers [13], and the related resources are referenced with hyperlinks (HATEOAS). As stated in Fielding's thesis [12], the uniform interface architecture is optimized for a "large-grain hypermedia data transfer", and it "degrades efficiency, since information is transferred in a standardized form rather than one which is specific to an application's needs". According to Fielding, the benefits of the uniform interface are as follows: (1) information hiding, as a client interacts only with a representation of resource; (2) decoupling of implementation and provided services allows evolvability; (3) simplified architecture; and (4) more visible interactions. GraphQL, on the other hand, does not provide a uniform interface similar to REST. There are three types of operations available: query for fetching data, mutation for modification of data, and subscriptions for receiving updates when data is updated [26]. These operations are

not bound to HTTP methods but are indicated by a keyword. Because GraphQL allows application-specific queries and only obtains desired values, it should be more efficient than REST.

### 3.6. Ease of Use

REST offers a uniform interface, and a user knows beforehand the possible operations on resources. Another benefit of REST is hyperlinks that allow referencing to a specific resource with URI, which is not possible with GraphQL because there is only one URI for the whole API. HATEOAS requires using hyperlinks as references to related resources. A drawback of REST is that fetching desired data often requires several requests. For example, with an OPC UA server, receiving information on referenced nodes requires one extra request per node (to overcome this, some solutions automatically fetch information on referenced nodes [1,5]).

The GraphQL approach for ease of use is the introspectable schema. It is a powerful feature and enables examination of the OPC UA server structure. In a query, the fields to be fetched are specified, and the shape of the response is similar to the request, making the response predictable for the user. In addition, compared to REST, it is possible to obtain information from related objects within nested requests [26].

Brito and Valente [22] conducted user tests on GraphQL and REST, which indicated that even with previous experience on REST, making queries was faster with GraphQL. In addition, GraphQL was considered easier to use because of the GraphiQL tool, which allows auto-complete and error detection of queries based on the schema, and more understandable syntax. Ease of use is an important factor when new technologies and shop-floor data are adopted. It leads to faster deployment and accelerates the development of new data-driven applications.

### 3.7. Ease of Development

Along with ease of use, ease of development determines the breadth of new services adoption. Ease of development is directly related to the costs of creating new applications and services. Both REST and GraphQL are back-end-agnostic and can use any data storage. In addition, both have numerous libraries to help development. However, as REST is older and more widely used, it offers better support for different web frameworks and programming languages.

With REST, the way in which objects in the data storage are mapped into resources and URIs must be defined. With GraphQL, resolver functions are used to define the handling of each field in the schema. GraphQL introduces a new paradigm to development, called the schema-first approach [23]. In this approach, the schema is already defined at the start of the project and helps to communicate the needs of developer teams to each other. In conclusion, there is no significant difference between REST and GraphQL regarding ease of development and both can be considered as being easy to develop.

### 3.8. Bandwidth Usage

This paper focuses on REST and GraphQL over HTTP because both are most commonly used with it, even though they are protocol-agnostic. Being a text-based protocol, HTTP has a large overhead; thus, fetching only a small amount of data per request is especially inefficient. For better efficiency, another protocol, such as CoAP, should be used instead of HTTP [30].

In RESTful approaches, each node is assigned a URI, which can contain, for example, *NameSpaceIndex* and *NodeId* [1]. Fetching multiple nodes and their values, such as the position of a pallet (x, y, z) [2], requires multiple requests. This can be circumvented by creating specific endpoints for commonly needed nodes and their values. The number of subsequent requests can also be reduced by prefetching information from referenced nodes [1], which increases the overhead if this information is not needed. On the other

hand, GraphQL allows aggregating multiple requests into a single query, and, for example, the position of the pallet could be fetched with only one request.

Theoretically, GraphQL has a lower bandwidth usage than REST because fewer requests are needed and only the specified data is returned. Brito, Mombach, and Valente [34] assessed the efficiency of GraphQL compared to REST in practice by migrating clients using GitHub and arXiv REST APIs to use the GraphQL version of the API. The number of queries did not significantly change, because clients already sent only one request to the API. However, the size of returned JSON documents decreased drastically: to the one-hundredth of the size returned by REST APIs.

### 3.9. Performance

The performance of REST and GraphQL are similar when a single resource is fetched as indicated in [24]. However, GraphQL gains an advantage on performance when multiple resources are fetched because it needs only one request whereas REST requires multiple requests. This was confirmed by performance analysis in [24]. In addition, measurements by Heredia et al. [33] showed that GraphQL offers a better performance compared to REST with smaller amount of resources used. However, the study did not specify the used queries; thus, the reason for better performance is unclear. The difference in performance can be reduced using HTTP pipelining [31], which allows sending multiple requests without waiting for a response between them. Caching, which is easier to implement with REST, also positively affects performance.

### 3.10. Information Model

The information model of OPC UA is a graph of connected nodes [2]. Therefore, it conforms well to a graph-like data model of GraphQL. With REST, a graph-like data model can be partly achieved by presenting hierarchies between objects within URIs and referencing to related resources with hyperlinks, following HATEOAS. However, URIs with long hierarchy chains are not considered to be good practice [14].

### 3.11. Other Features

GraphiQL is a browser-based integrated development environment (IDE) that provides several features: introspection of the schema, executing queries, code completion, syntax highlighting, and error warnings [23]. In an experiment by Brito and Valente [22], GraphiQL was one of the major factors which resulted in GraphQL being easier to use than REST. The GraphQL type system can also be queried without GraphiQL, enabling its introspection. Interactive tools, such as ReadMe (https://readme.com/, accessed on 13 March 2022), are also available to REST APIs, although they are not an official part of the architectural style. In addition, there is a standardization effort for describing REST APIs called OpenAPI Specification (OAS) (https://www.openapis.org/, accessed on 13 March 2022).

GraphQL supports deprecating fields and adding new ones without the need for API versioning [34]. According to Fielding [35], if the HATEOAS principle is followed, REST does not require versioning as "controls have to be learned on the fly". Unfortunately, the HATEOAS principle is often violated by so-called REST APIs [13], and versioning might be needed. The version of the API can be indicated in the URL, header, media type, or parameter of a query string [14].

### 4. Measurement Setup

To compare the request execution times of REST and GraphQL, measurements with a simple OPC UA server were conducted. The measurement setup (Figure 2) consisted of a laptop acting as a client and capturing network traffic with Wireshark (https://www.wireshark.org/, accessed on 13 March 2022), a second laptop hosting the REST/GraphQL interface server, two Raspberry Pi 4 Model B 2 GB hosting OPC UA server and cache server, and a switch mirroring the network traffic to the first laptop. Laptops had Intel i5-1135G7

and i5-7200U processors for the client and interface server, respectively, and 16 GB and 8 GB of RAM.

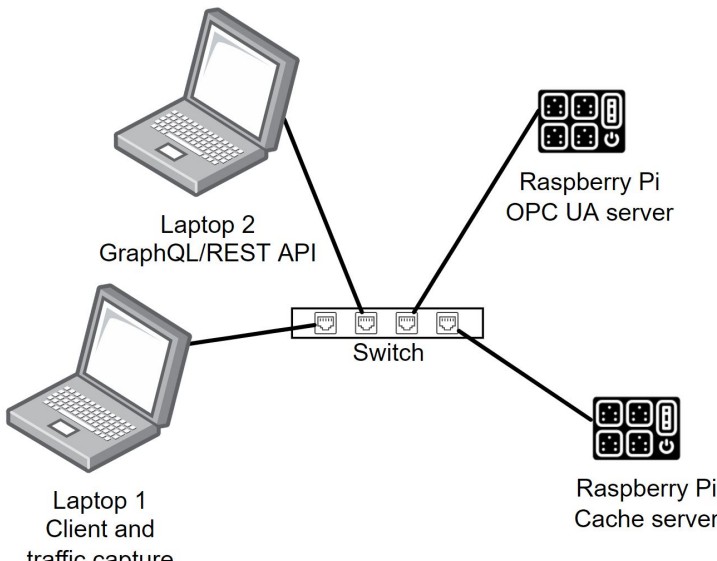

**Figure 2.** The setup for measuring the request execution times to read and write values to the OPC UA server.

For the REST interface, we used the implementation presented by Cavalieri et al. in [1], which is available from [18], and for GraphQL interface solution by Hietala et al. introduced in [8], available from [27]. In order to have similar authentication methods for both interfaces, token-based authentication was added to the GraphQL API. Authentication uses JSON Web Tokens and each request must contain this token in the *Authorization* request header. To enable authentication, the web framework of the GraphQL server was changed to FastAPI, which is based on Starlette. The REST client used multi-threading with four threads, which allows simultaneously sending multiple requests to the interface. OPC UA server was implemented with Python FreeOpcUa library [36], and for the communication with the server TCP/IP stack with OPC UA binary protocol was used.

Each experiment was conducted 1000 times, and the total request execution times were recorded by a client using Python time.time() function. To investigate the effect of connection establishment on the request execution time, the experiments were performed both without a connection as well as with the client/interface already connected to the OPC UA server. The experiments on request execution times were as follows:

1. Reading a single value from the OPC UA server.
2. Writing a single value to the OPC UA server.
3. Reading 50 values from the OPC UA server.
4. Writing 50 values to the OPC UA server.

With the REST interface, reading data was also performed via cache server. The cache server responded either with the cached response (steps 1 and 6 in Figure 3) or, in the second experiment, forwarded the query to the REST server, which then read data from the OPC UA server (steps 1–6). The first measurement was excluded from the cached value results as it fetched the value that was returned in the subsequent requests. The first measurement was also excluded with other REST measurements to ensure connection with the OPC UA server. The selected test cases reflect reading or writing a simple status of a machine via an OPC UA server, such as "system ok" and reading or writing all variables to the server. For example, the industrial machine in our laboratory had approximately 50 meaningful variables after duplicates are removed.

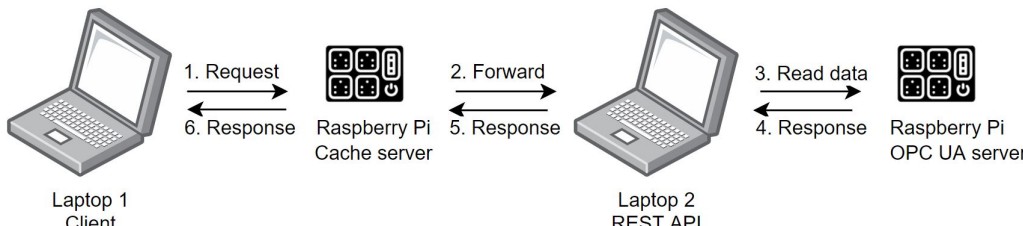

**Figure 3.** The effect of cache server for REST performance was examined by measuring read times when cached result was returned and when the value needed to be fetched from the OPC UA server.

The network traffic was captured to distinguish the processing of interfaces and communication with the OPC UA server from the total execution time. For calculating these shares, the timestamps from Wireshark were used to calculate averages of the processing time and the communication duration with OPC UA. These averages were then subtracted from the request execution time recorded by the client. In addition, the TCP payload (size of headers were excluded) of requests and responses were measured with Wireshark by running the client, interface, and OPC UA server on the same computer. The query formatting, that is, the use of spaces, tabulators, and line breaks, significantly affects the size of requests. Thus, the GraphQL queries were run on an easy-to-read format (which was also used when measuring request execution times) and minified form, in which only necessary spaces were present. REST interface, on the other hand, returned an *x-token* header, which to the authors' best knowledge presented the freshest token. The token was not updated after each request and including the token in each response was considered unnecessary. Therefore, the sizes of the responses were also measured without the *x-token*.

## 5. Results

The request execution times are summarized in Table 2. The measurements show that using OPC UA directly, without an intermediary interface, is significantly faster, even when the client needs to first connect to the OPC UA server. However, reading a single cached value is faster than reading a fresh value from OPC UA server. Writing to OPC UA server is slightly faster than reading with a single value, but reading becomes quicker when the number of values increases.

If web interfaces are already connected to the OPC UA server, the median of reading a single value is 13% slower and writing is 54% slower with GraphQL. With multiple values, GraphQL outperforms REST with a clear margin: the medians of reading and writing values are 268% and 24% slower, respectively. This is expected because the REST interface requires one request per each value, whereas one GraphQL query can be used to interact with several values. Using cache server significantly reduces read times with REST server, and fetching a cached value is even faster than connecting to a OPC UA server and reading fresh value. However, use of cache server increases latency if cached value cannot be used and the REST server needs to read data from OPC UA server. The standard deviation of measurements was relatively high, especially with the REST interface, and the measurements in which the REST server first needed to connect to the OPC UA server were excluded from the table. The request execution times for the disconnected REST server were over 1 s for reading/writing a single value and over 2.5 s for reading/writing 50 values.

**Table 2.** The request execution times in milliseconds for reading and writing values to the OPC UA server. SD = standard deviation. Disconnected and connected indicate if the OPC UA session was established before the request was made.

|  | Min | Max | Mean | Median | SD |
|---|---|---|---|---|---|
| **Read 1 value** | | | | | |
| **Connected** | | | | | |
| OPC UA | 5.3 | 21.2 | 7.0 | 7.1 | 1.1 |
| REST | 47.8 | 98.6 | 62.9 | 63.6 | 4.9 |
| GraphQL | 56.9 | 148.9 | 73.0 | 71.9 | 8.2 |
| **Disconnected** | | | | | |
| OPC UA | 27.6 | 75.1 | 36.3 | 36.3 | 4.1 |
| GraphQL | 78.8 | 199.1 | 97.3 | 96.3 | 10.7 |
| **Cache** | | | | | |
| REST cached | 8.9 | 19.6 | 13.0 | 13.2 | 1.1 |
| REST via cache | 63.1 | 112.1 | 77.1 | 77.9 | 4.7 |
| **Write 1 value** | | | | | |
| **Connected** | | | | | |
| OPC UA | 5.4 | 17.3 | 7.3 | 7.3 | 1.0 |
| REST | 31.4 | 87.8 | 42.0 | 42.2 | 3.6 |
| GraphQL | 55.3 | 96.9 | 67.0 | 65.1 | 5.1 |
| **Disconnected** | | | | | |
| OPC UA | 28.4 | 70.4 | 36.7 | 36.5 | 4.1 |
| GraphQL | 75.1 | 186.6 | 90.6 | 90.0 | 10.5 |
| **Read 50 values** | | | | | |
| **Connected** | | | | | |
| OPC UA | 38.8 | 47.6 | 43.6 | 43.7 | 1.6 |
| REST | 856.0 | 1057.6 | 915.9 | 906.1 | 35.4 |
| GraphQL | 177.7 | 414.6 | 248.9 | 246.3 | 34.5 |
| **Disconnected** | | | | | |
| OPC UA | 32.9 | 94.1 | 62.9 | 70.6 | 13.7 |
| GraphQL | 186.7 | 441.4 | 270.7 | 269.9 | 38.9 |
| **Cache** | | | | | |
| REST cached | 206.6 | 1328.7 | 284.6 | 297.4 | 79.6 |
| REST via cache | 927.0 | 1258.2 | 1095.7 | 1096.0 | 60.3 |
| **Write 50 values** | | | | | |
| **Connected** | | | | | |
| OCP UA | 44.1 | 51.8 | 48.1 | 48.3 | 1.4 |
| REST | 560.4 | 1470.9 | 615.5 | 602.2 | 65.6 |
| GraphQL | 316.0 | 788.9 | 482.4 | 486.7 | 60.7 |
| **Disconnected** | | | | | |
| OPC UA | 34.6 | 100.5 | 59.7 | 49.4 | 15.0 |
| GraphQL | 403.1 | 1108.0 | 574.8 | 587.4 | 65.7 |

Figure 4 shows the amount of time required by the processing, communication with OPC UA server, and transport of messages between client and interface from the total request execution time. The processing times are higher with the GraphQL interface because the query must be parsed to corresponding OPC UA service requests. REST requires more time to communicate with the OPC UA server due to extra service requests, such as a browse request to obtain information on referenced nodes. It was also noted that significantly more time is required for the REST interface to connect to the OPC UA server due to several extra messages being sent, such as handshakes. Moreover, fetching authorization token was much slower with the REST interface compared to GraphQL. Nevertheless, these drawbacks are related to the implementation of the REST interface, rather than REST itself.

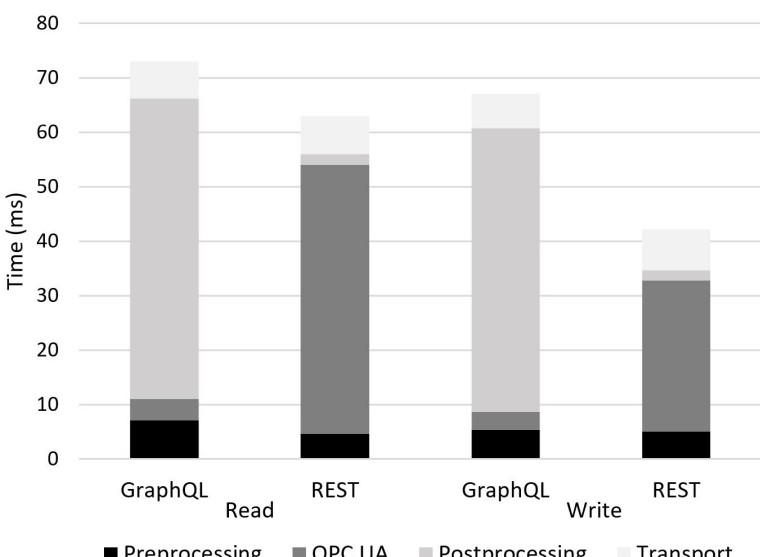

**Figure 4.** The share of processing times, times to read/write data from/to OPC UA server, and transport times from the total request execution times.

Communicating directly with the OPC UA server yields the lowest TCP payloads. When reading or writing a single value, the difference between REST and GraphQL is relatively small (Figure 5). Yet, with a larger number of variables, the payload is significantly smaller using GraphQL (Figure 6). This is because REST needs to send multiple requests, whereas GraphQL needs only one request. It can be also seen that minifying the request significantly reduced its size with GraphQL. In addition, leaving the unnecessary *x-token* out reduced the REST response size to approximately half. No authorization was used with OPC UA, which reduced the payload size.

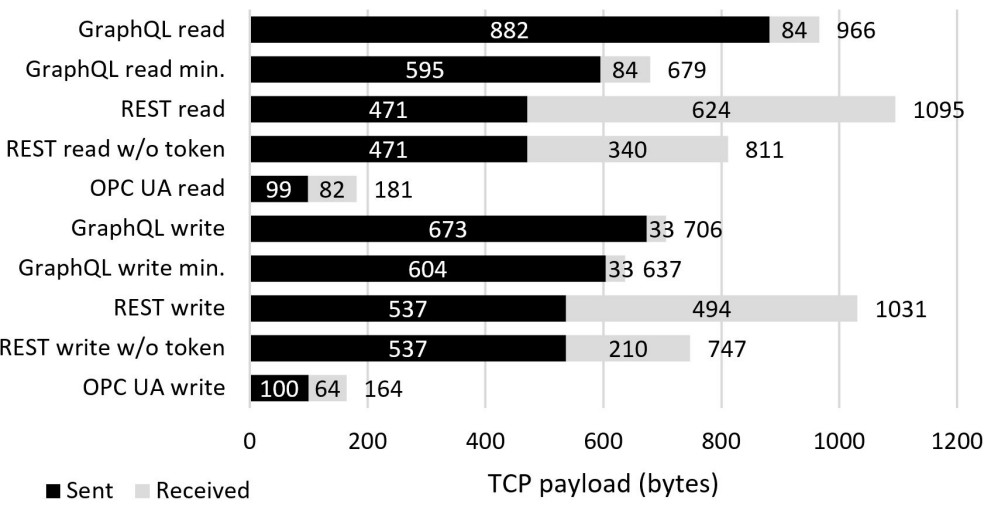

**Figure 5.** The bytes sent and received (TCP payload) by a client when reading/writing a single value. min. = minified, w/o = without.

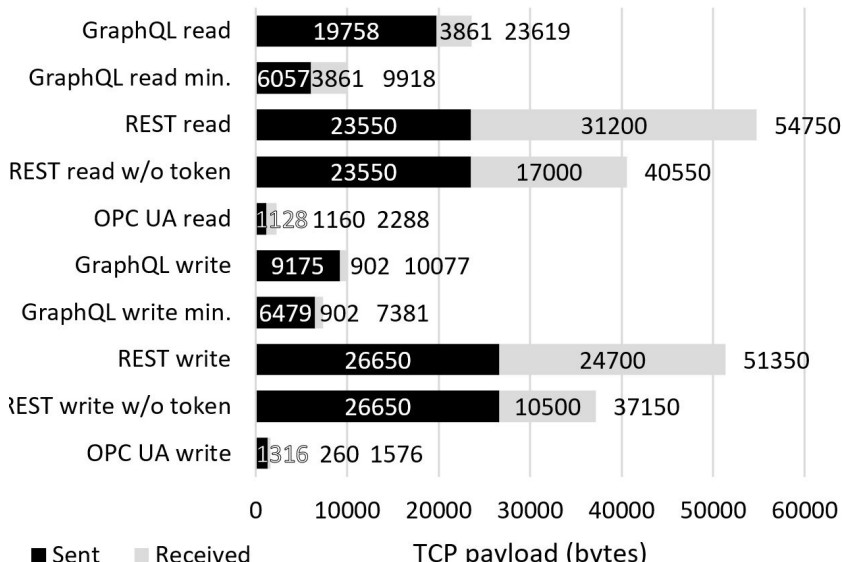

**Figure 6.** The bytes sent and received (TCP payload) by a client when reading/writing a value 50 times. min. = minified, w/o = without.

## 6. Discussion

This paper examined bringing web interfaces, REST and GraphQL, to an industrial domain as an additional interface to OPC UA servers. The qualitative analysis showed that GraphQL offers more favorable features than REST. In addition, the performance measurements and bandwidth use preferred GraphQL over REST. However, OPC UA binary over TCP outperformed both web-based interfaces.

The hypothesis of the paper was that REST and GraphQL perform similarly with a single value, but with multiple values, the nested queries and ability to specify the fetched data favor GraphQL. This hypothesis was otherwise supported by the measurements, but writing a single values was significantly faster with REST. In addition, it was hypothesized that direct communication with OPC UA would be slower than communication with web interfaces if the client first needs to connect to the OPC UA server and the interfaces are already connected to the server because of the several handshakes required for establishing OPC UA connection (Figure 1). However, the measurements showed that OPC UA is faster in each measurement case, except if a cache server is used. Using a cache server improves the performance significantly, but it might not be possible to use cached values.

Threats to the validity of the measurements were posed by a relatively high standard deviation. The high deviation can be explained by the communication over the network and the non-consistent operation of the REST interface. For a more fair comparison between OPC UA and web interfaces, OPC UA over HTTP with JSON payload should have been used. Nevertheless, based on the authors' experience, the OPC UA binary over TCP stack is a more commonly used method for communication and, thus, was selected for the comparison. The effect of authentication methods could also be examined in future research.

Even though GraphQL and REST increase the latency, both can be considered suitable for at least monitoring purposes. Easy data collection from the manufacturing system enables analyzing the manufacturing process and improving the process based on this information. In addition, web interfaces can be used to send commands to industrial machines, for example, ask a crane to drive to a certain position. In the previous work [8], the authors used GraphQL to control an industrial crane via the web user interface. GraphQL was also used to monitor and control the crane with augmented reality application [4]. However, web APIs are not suitable for real-time control, in which messages are guaranteed to arrive within a certain time frame.

It is recommended that GraphQL API is offered as an additional interface for the OPC UA server. This additional interface would allow accessing data and introspection

of the OPC UA server from a browser using the GraphiQL tool. GraphQL interface would also make accessing the OPC UA server easier for the developers and improve the interoperability of the OPC UA with standard web technologies. This would enable the development of advanced data-driven applications and pave the way towards Industry 4.0 in which cyber–physical systems communicate with each other over a network.

Proposed future work includes a comparison of web interfaces to OPC UA over HTTP. In addition, it should be examined if OPC UA servers could be made available to the Internet via web interfaces. The web interfaces would act as a gateway between the private network inside a factory and the public Internet. Finally, more use-case-driven tests could be conducted to assess the performance of REST and GraphQL.

## 7. Conclusions

This paper compared REST and GraphQL interfaces for OPC UA servers in the industrial domain. GraphQL was considered to be easier to use because it provides the GraphiQL tool, which allows introspection of OPC UA servers and assists in writing queries. Both REST and GraphQL use HTTP protocol making OPC UA interoperable with common Web technologies. Therefore, the OPC UA server can be accessed without an OPC UA specific client.

GraphQL was considerably faster than REST when multiple values were read or written, and REST offered a better performance when a single value was written. However, to achieve the best performance, OPC UA should be directly used without an intermediate interface server. If extremely high scalability is required, the use of REST might be justified since it allows load-balancing and more versatile caching. In all other use cases, this paper recommends using GraphQL over REST because it offers better performance with multiple values, lower bandwidth usage, and ease of use. This paper also recommends providing the GraphQL interface parallel with direct access to the OPC UA server to obtain the best of both solutions, such as interoperability, developer-friendliness, and performance.

**Author Contributions:** Conceptualization, R.A.-L., J.A. and K.T.; methodology, R.A.-L., J.M. and J.A.; software, R.A.-L., J.M., J.H. and H.L.; validation, R.A.-L., J.A. and K.T.; formal analysis, R.A.-L., J.M. and J.A.; investigation, R.A.-L. and J.M.; resources, K.T.; data curation, R.A.-L., J.M. and J.H.; writing—original draft preparation, R.A.-L. and J.A.; writing—review and editing, R.A.-L., J.M., J.A. and K.T.; visualization, R.A.-L. and J.M.; supervision, H.L. and K.T.; project administration, K.T.; funding acquisition, H.L. and K.T. All authors have read and agreed to the published version of the manuscript.

**Funding:** This research was funded by the Business Finland under Grant 3508/31/2019 and ITEA 3 Call 5 MACHINAIDE.

**Institutional Review Board Statement:** Not applicable.

**Informed Consent Statement:** Not applicable.

**Acknowledgments:** R. Ala-Laurinaho would like to thank Tekniikan edistämissäätiö.

**Conflicts of Interest:** The authors declare no conflict of interest. The funders had no role in the design of the study; in the collection, analyses, or interpretation of data; in the writing of the manuscript, or in the decision to publish the results.

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
