# Peer review of "Comparison of REST and GraphQL Interfaces for OPC UA"

_computers, doi:10.3390/computers11050065_

Round 1

Reviewer 1 Report

The paper presents in details a comparison between two types of Application Programming Interfaces (APIs) used in industrial communication: Representational State Transfer (REST) and GraphQL. I appreciated the extended survey regarding these two solutions (i.e. the related work paragraphs), whilst the testbed scenario involved in an efficient manner the existing software publicly available in github.com (previously developed by the authors or by other researchers).

Some suggestions to improve the paper:

  1. The acronyms are repeated too many times in the same phrases. OPC UA (Open Platform Communications Unified Architecture) is the most relevant example. E.g. in rows 124-125 the readers understand that you refer to this architecture. So you could better say: "However making it fully REST compliant has proven challenging because this architecture is originally a stateful protocol...". See also RESTful in row 119 etc.
  2. Some references within the text are missing (see [?] at rows 197, 268, 309 etc.)
  3. It is better to move the links to GitHub into the references (e.g. rows 218-219, 224 etc.)
  4. Table 2 in page 11 is quoted after its first appearence. This could be easily solved by moving rows 452-456 before Table 2.
  5. Some acronyms not explained. (e.g. URI in row 106).
  6. Some typos: e.g. row 113 ("separated" instead of "separate"), row 510 ("authors of [6] used ... instead "authors [6], authors used ...").  
  7. Keywords in alphabetical order
  8. The authors avoided to quote some of their previous publications or presentations regarding the same topic, containing better explanations that are missing in this paper. I refer mainly to some figures: e.g. Figure 9 , page 24 from J. Hietala, "Real-time two-way data transfer with a Digital Twin via web interface" and similar others. 

Author Response

The authors would like to thank the reviewer for the comments. Below are detailed answers to each of the concerns:

Concern #1: The acronyms are repeated too many times in the same phrases. OPC UA (Open Platform Communications Unified Architecture) is the most relevant example. E.g. in rows 124-125 the readers understand that you refer to this architecture. So you could better say: "However making it fully REST compliant has proven challenging because this architecture is originally a stateful protocol...". See also RESTful in row 119 etc.

Author response: The authors would like to thank for the comment, as reducing acronyms makes the text flow better. However, sometimes using acronyms is required to make the text explicit.

Author action: Authors replaced some of the acronyms by changing wording.

Concern #2: Some references within the text are missing (see [?] at rows 197, 268, 309 etc.)

Author response: The authors would like to thank for the comment. All [?] are the same reference to GraphQL specification (ref number 23). "GraphQL contributors. GraphQL specification October 2021. http://spec.graphql.org/October2021/, 2021"

Author action: The broken reference was replaced, and also other references were checked again.

Concern #3: It is better to move the links to GitHub into the references (e.g. rows 218-219, 224 etc.)

Author response: The authors would like to thank for the comment and agree with the reviewer. Moving links to GitHub to references increases clarity of the paper.

Author action: GitHub links were moved from in-text to the references section.

Concern #4: Table 2 in page 11 is quoted after its first appearence. This could be easily solved by moving rows 452-456 before Table 2.

Author response: The authors would like to thank for the comment.

Author action: The table was moved as the reviewer suggested.

Concern #5: Some acronyms not explained. (e.g. URI in row 106).

Author response: The authors would like to thank for the comment. Some of the most commonly used acronyms were not initially opened. However, we agree that these acronyms should be opened.

Author action: The authors double checked all acronyms and opened them.

Concern #6: Some typos: e.g. row 113 ("separated" instead of "separate"), row 510 ("authors of [6] used ... instead "authors [6], authors used ...").

Author response: The authors would like to thank the reviewer for pointing out these typos. Typos reduce the credibility of the paper.

Author action: The authors removed the typos reported by the reviewer and read the paper carefully again.

Concern #7: Keywords in alphabetical order

Author response: The authors would like to thank the reviewer for the comment.

Author action: The authors arranged the keywords in alphabetical order.

Concern #8: The authors avoided to quote some of their previous publications or presentations regarding the same topic, containing better explanations that are missing in this paper. I refer mainly to some figures: e.g. Figure 9 , page 24 from J. Hietala, "Real-time two-way data transfer with a Digital Twin via web interface" and similar others.

Author response: The authors would like to thank the reviewer for the comment. Hietala's master's thesis was not originally referenced since there is a peer-reviewed conference paper based on the thesis available. Figure 1 in the current paper is essentially the same figure as Figure 9 in the thesis.

Author action: Added reference to Hietala's master's thesis and clarified the explanation of Figure 1.

Reviewer 2 Report

The authors present a technical comparison between REST and GraphQL. Although the authors presented several references with respect to those web interfaces, there is no contribution. The paper requires additional work such as: providing more results since only read/write time are including. What about delay, cache performance, computacional cost, etc. Please, add those results by performing more experimental tests. Also, there are few bad references [?]. Furthermore, I'm wondering why do you compare those web interfaces when OPC UA server is much more efficient. You should explore that research. 

Author Response

The authors would like to thank the reviewer for the comments. Below are detailed answers to each of the concerns:

Concern #1: The authors present a technical comparison between REST and GraphQL. Although the authors presented several references with respect to those web interfaces, there is no contribution.

Author response: The authors would like to thank the reviewer for the comment. Authors strongly disagree that there is no contribution. The use of HTTP-based Internet-native interfaces is crucial to the progress towards Industry 4.0, in which several factories and machines communicate with each other over the public Internet. HTTP-based protocols allow the development of industrial applications with various programming languages and technologies that is not possible using OPC UA (See also response to Concern #4). In addition, there is no scientific literature that compares REST and GraphQL for OPC UA, and literature about using REST and GraphQL in the industrial domain is scarce. This comparison paper is necessary to allow researchers and also industry representatives to make informed decisions on how they should provide access to their OPC UA servers over the public Internet. We respectfully note the reviewer has not shown an existing contribution nullifying our novelty claim.

Author action: The authors added better justification of the importance of the research to the introduction section

Concern #2: The paper requires additional work such as: providing more results since only read/write time are including. What about delay, cache performance, computacional cost, etc. Please, add those results by performing more experimental tests.

Author response: The authors would like to thank the reviewer for the comment. From the client's perspective (that is the focus of this paper), the reading and writing times are the most relevant measures of performance. Measuring the cache was a good proposal as the cache significantly reduces read times. We conducted additional tests using a cache server to measure how it affects read performance of REST server. We also repeated measurements with a higher number of samples: reading and writing tests were now performed 1000 times. The computational cost related to a specific interface is not clearly defined as the interface does not specify how the response should be processed and data fetched from the database or, in this case, OPC UA server. It is true that a GraphQL interface needs to do some additional processing for parsing the query and constructing the response compared to REST. The amount of processing is highly dependent on the query and the generalization of computational cost is not meaningful.

Author action: The authors conducted additional measurements of REST server performance with caching.

Concern #3: There are few bad references [?].

Author response: The authors would like to thank for the comment. All [?] are the same reference to GraphQL specification (ref number 23). "GraphQL contributors. GraphQL specification October 2021. http://spec.graphql.org/October2021/, 2021"

Author action: The broken reference was replaced, and also other references were checked again.

Concern #4: Furthermore, I'm wondering why do you compare those web interfaces when OPC UA server is much more efficient. You should explore that research.

Author response: The authors would like to thank for the comment. It is inevitable that an additional interface on top of another interface increase latency. However, the purpose of the additional interface is not to improve the performance but the interoperability and accessibility of OPC UA servers. The interoperability of OPC UA when developing application is still a huge problem. For example, when an AR application was developed for an industrial crane (https://doi.org/10.3390/app11209480), there was no easy way to communicate with OPC UA using C# programming language. Therefore, GraphQL interface has to be used to access the crane OPC UA server. The same problem has also occurred with other programming languages, such as MicroPython. As HTTP is the internet protocol, programming languages provide almost always support it. Therefore, if HTTP-based interfaces, such as REST or GraphQL, can be provided to access OPC UA server, it enables the development of various applications. (There is also HTTP-based OPC UA available, but it might not be supported by the server.)

Author action: The authors added better justification of the importance of the research to the introduction section.

Reviewer 3 Report

In this paper, characteristics and performance of REST and GraphQL interfaces for OPC UA in industrial communication domain were contrastive analyzed, and measurements on data reading and writing were conducted. Overall, I found the work was interesting and of good importance. The results of the study have practical reference value for improving interoperability between different devices in industrial domain. However, there are still some imperfections in the work, and I hope that my comments would be useful for improving the quality of the paper. Some of detailed comments are as follows:

  1. Although the OPC UA, REST and Graph QL interfaces are described in detail separately, a brief supplementary instruction to the relationship between them is recommended to further improve the readability of the article.
  2. There are some formatting errors that need to be modified. Such as “the specifcation [? ] are as follows”, “The GraphQL specifcation [? ]”, “when data is updated [? ]”, “within nested requests [? ]”, “, authors [6], authors used” and so on.

Author Response

The authors would like to thank the reviewer for the comments. Below are detailed answers to your concerns:

Concern #1: Although the OPC UA, REST and Graph QL interfaces are described in detail separately, a brief supplementary instruction to the relationship between them is recommended to further improve the readability of the article.

Author response: The authors would like to thank the reviewer for the comment. The relationship is currently quite hidden in the text. Basically, major difference from the client point of view is the information model of the interfaces. There is also Table 1 that summarizes the properties of REST and GraphQL in industrial communication.

Author action: The authors briefly state the relationship between REST and GraphQL in section 2.4.

Concern #2: There are some formatting errors that need to be modified. Such as “the specifcation [? ] are as follows”, “The GraphQL specifcation [? ]”, “when data is updated [? ]”, “within nested requests [? ]”, “, authors [6], authors used” and so on.

Author response: The authors would like to thank for pointing out these formatting errors. All [?] are the same reference to GraphQL specification (ref number 23). "GraphQL contributors. GraphQL specification October 2021. http://spec.graphql.org/October2021/, 2021".

Author action: The broken reference was replaced, and also other references were checked again. The paper was rechecked for grammatical errors.

Round 2

Reviewer 1 Report

I agree with the changes.